# Marine Science Can Contribute to the Search for Extra-Terrestrial Life

**DOI:** 10.3390/life14060676

**Published:** 2024-05-24

**Authors:** Jacopo Aguzzi, Javier Cuadros, Lewis Dartnell, Corrado Costa, Simona Violino, Loredana Canfora, Roberto Danovaro, Nathan Jack Robinson, Donato Giovannelli, Sascha Flögel, Sergio Stefanni, Damianos Chatzievangelou, Simone Marini, Giacomo Picardi, Bernard Foing

**Affiliations:** 1Instituto de Ciencias del Mar (ICM)—CSIC, 08003 Barcelona, Spain; nathan@icm.csic.es (N.J.R.); damianos@icm.csic.es (D.C.); gpicardi@icm.csic.es (G.P.); 2Stazione Zoologica Anton Dohrn, Villa Comunale, 80121 Naples, Italy; sergio.stefanni@szn.it (S.S.); simone.marini@cnr.it (S.M.); 3Natural History Museum, Cromwell Road, London SW7 5D, UK; j.cuadros@nhm.ac.uk; 4School of Life Sciences, University of Westminster, 115 New Cavendish St, London W1W 6UW, UK; lewis.dartnell@gmail.com; 5Consiglio per la Ricerca in Agricoltura e l’Analisi Dell’Economia Agraria—Centro di Ricerca Ingegneria e Trasformazioni Agroalimentari, 00015 Monterotondo, Italy; corrado.costa@crea.gov.it (C.C.); simona.violino@crea.gov.it (S.V.); 6Consiglio per la Ricerca in Agricoltura e l’Analisi dell’economia Agraria—Centro di Ricerca Agricoltura e Ambiente, 00182 Roma, Italy; loredana.canfora@crea.gov.it; 7Department of Life and Environmental Sciences, Polytechnic University of Marcs (UNIVPM), 60131 Ancona, Italy; r.danovaro@univpm.it; 8Department of Biology, University of Naples Federico II, 80138 Naples, Italy; donato.giovannelli@gmail.com; 9National Research Council—Institute of Marine Biological Resources and Biotechnologies (CNR-IRBIM), 60125 Ancona, Italy; 10Department of Marine and Coastal Science, Rutgers University, New Brunswick, NJ 08901, USA; 11Marine Chemistry, Geochemistry Department—Woods Hole Oceanographic Institution, Falmouth, MA 02543, USA; 12Earth-Life Science Institute, Tokyo Institute of Technology, Tokyo 152-8552, Japan; 13GEOMAR Helmholtz Centre for Ocean Research, 24106 Kiel, Germany; sfloegel@geomar.de; 14Institute of Marine Sciences, National Research Council of Italy (CNR-ISMAR), 19032 La Spezia, Italy; 15Faculty of Earth and Life Sciences, Vrije Universiteit Amsterdam, De Boelelaan 1081-1087, 1081 HV Amsterdam, The Netherlands; foing@strw.leidenuniv.nl

**Keywords:** term-map clusters, deep-sea, icy moons, habitability, extremophiles, extraterrestrial intelligence

## Abstract

Life on our planet likely evolved in the ocean, and thus exo-oceans are key habitats to search for extraterrestrial life. We conducted a data-driven bibliographic survey on the astrobiology literature to identify emerging research trends with marine science for future synergies in the exploration for extraterrestrial life in exo-oceans. Based on search queries, we identified 2592 published items since 1963. The current literature falls into three major groups of terms focusing on (1) the search for life on Mars, (2) astrobiology within our Solar System with reference to icy moons and their exo-oceans, and (3) astronomical and biological parameters for planetary habitability. We also identified that the most prominent research keywords form three key-groups focusing on (1) using terrestrial environments as proxies for Martian environments, centred on extremophiles and biosignatures, (2) habitable zones outside of “Goldilocks” orbital ranges, centred on ice planets, and (3) the atmosphere, magnetic field, and geology in relation to planets’ habitable conditions, centred on water-based oceans.

## 1. Introduction

Several planets and moons in our Solar System host large bodies of water, termed exo-oceans [1], that are often analogous to those of Earth, albeit below thick ice shells [2]. Marine environments may exist on Saturn or Jupiter’s moons, such as Enceladus, Europa, Callisto and Ganymede [3,4,5,6,7,8,9,10,11,12]. Mars once had large lakes and possibly seas and still has subsurface liquid water at its South Pole, where a briny solution is keeping water liquid beneath the ice shell [13,14,15]. Titan is also hypothesized to contain water, maintained in a liquid state by ammonia, beneath its water-ice crust on which there are rivers and seas of liquid methane and ethane [16,17,18].

More extraterrestrial water-ocean environments are likely to be discovered on exo-planets over the next decade [19,20,21]. Inhabitable oceans have been hypothesized to exist on planets orbiting stars beyond the previously considered habitable range [22,23,24,25]. Habitability near M-red dwarf stars is problematic because they typically emit lethal flares [26], even though in some cases this may be counteracted by a planet’s magnetic field [27], ozone shields [28], and/or water bodies [29]. Beyond the traditionally defined stellar habitable zone, life could also exist in geothermally or tidal friction-heated liquid oceans below thick ice shells [2].

The Earth’s deep sea is a vast biome devoid of sunlight, where high-pressure and different cyclic hydrodynamic phenomena occur at tidal, inertial, and seasonal scales within the water column and the seabed [30,31]. In the benthic realm, communities of chemosynthetic bacteria and multicellular organisms thrive in areas of high geothermal activity [32]. It has been hypothesized that life on Earth may have originated around marine hydrothermal vents with considerable geothermal activity [33], as these areas represent a habitat that fulfils the requirements of constant energy and essential chemical element supply, alongside other favorable conditions [34,35]. Precipitation of vesiculated metal oxides and/or sulphones at vents has been considered to create compartments acting as primitive cells, able to trap organic molecules, and reach concentration potentials across the walls and catalyze reactions [36]. Similar ecosystems can also be found in exo-oceans on moons and planets with geothermally active mantles and/or cores [37,38]. These systems may present environmental conditions analogous to those of Earth’s oceans [39,40,41,42,43,44], where water–rock interactions might provide nutrients and trace elements together with thermodynamic disequilibria. Life could also have arisen and evolved there, as speculated, for example, for Enceladus and Europa, which have the same age as the Earth [12]. Extraterrestrial marine organisms may have developed to the multicellular stage, and they could even display some level of morphological convergence with terrestrial analogues, following the principle that morphological similarity in non-phylogenetically related species can be predicted from common environmental conditioning factors [45].

The hypothesis that uni- or multicellular extraterrestrial organisms may have evolved around terrestrial-like hydrothermal vents could help us to conceptually explore exo-oceans for life signatures. How we imagine these putative extraterrestrial life forms (i.e., uni- or multicellular, photoautotrophs, chemo-autotrophs or heterotrophs [46,47,48]) will influence the approach we use for their identification. Since the resounding successes of the Voyager 1 and 2 probes, Galileo and Cassini-Huygens, between 1979 and 2017, future missions to Enceladus, Europa, Ganymede, Titan, and Ceres are now being conceived [2,12], based on robotic platforms for orbital surveying, ice-shell drilling, pelagic navigation and rocky-core seabed sampling, all endowed with different spectrophotometric and imaging sensors for molecular and microbial detection [49,50,51,52,53,54,55,56,57,58,59]. Specifically, investigation of the Earth’s deep sea is developing technology for the operation of novel autonomous vehicles and their payloads devoted to environmental exploration and ecological monitoring [60]. Driven by technological advancements and the fundamental interest in submarine life, new geomorphological habitats and associated ecosystems are constantly being discovered at continental margins and in abyssal realms [31]. The potential of submarine exploration is illustrated by the fact that hydrothermal vents, a revolutionary discovery, were found in 1977, long after humanity had already landed on the Moon.

This technological innovation is providing new methods for the detection and monitoring of life with potential applications to astrobiological research [60,61]. For instance, the development of vessel-teleoperated remotely operated vehicles (ROVs) completely revolutionized research in the deep-sea water column [62,63]. Similarly, both macro- and microscopic imaging, in situ spectroscopy, multibeam sonar imaging, and passive acoustic sensors (hydrophones) are playing an increasing role in the investigation of life [64]. More recently, new life-detecting capabilities have been implemented through the development of eco-genomic sensors, based on environmental DNA (eDNA), and delivering information on the presence of species across a wide range of ecological sizes (i.e., from bacteria to whales), complementing imaging or acoustics [30,65]. In the case of exo-oceans, such developments could be used for the detection of DNA-based life-forms. In deep-sea research, the mounting of these image- and molecular-based sensors not only on ROVs, but also on other autonomous or remotely operated robots (e.g., Autonomous Underwater Vehicles-AUVs, rovers and crawlers), enables the assessment of the presence of species and their relationship with the environment, by coupling biological data acquisition with synchronous oceanographic and geochemical surveying [61]. Specifically, for each image/sound or molecular or eDNA record, one can establish the associated environmental status from measurements with geochemical and oceanographic sensors [61,65]. Recently, off-the-shelf platforms with such multiparametric sensors have been described in relation to the exploration of possible marine life on Enceladus and Europa [61,66,67].

Bibliometric analysis tools such as VOSviewer [68] and CiteSpace represent valuable methods for exploring and analyzing large volumes of scientific data [69], manifesting astrobiology research fields and shedding light on emerging areas. Previous efforts in the analysis of astrobiology research trends with bibliometric methods [70] have related to authors, collaborating networks and research domains; [71] to emerging research topics and their evolution from three flagship journals; and [72] to biotechnological applications for biomining and bioleaching in long-term human space exploration. We approach our bibliographic investigation of exo-life from the perspective of connections with biological marine science. The identification of emerging astrobiological research areas and their relationship with marine science could provide a dialogue framework to promote strategies for the discovery and monitoring of uni- and multicellular extraterrestrial life under ‘marine-like’ conditions. We conducted a data-driven and quantitative bibliographic survey on astrobiology literature from its beginning (by combining VOSviewer and CiteSpace bibliometric analyses) to identify developing paradigms and research trends generally and, more specifically, potential connections with marine science, for future synergies in the exploration for extraterrestrial life in exo-oceans.

## 2. Materials and Methods

### 2.1. Bibliographical Query to Identify “Terms” of Interest

We conducted a preliminary screening of available scientific peer-reviewed literature in the SCOPUS database (Elsevier, Netherlands), following similar methodological approaches previously tuned in Costa et al. [73]. Although new statistical approaches exist to monitor the overall status of various research topics, one emerging method is scientific mapping. This approach allows for a review of research and is designed to synthesize patterns of knowledge production within a discipline, as opposed to synthesizing substantive findings. It is an interdisciplinary field emerging from traditional library information science in the areas of scientometrics, citation analysis, and computer science in the subareas of information visualization, visual analysis, data mining, and knowledge discovery [73].

The SCOPUS database was chosen because it contains a wide selection of scientific literature in marine and space sciences, equivalent to the ISI Web of Science.

For the definitive query, we entered several different combinations of astrobiology- and marine science-related keywords aiming at including in our search a large and consistent corpus of articles to allow for a solid statistical analysis. The search was tailored to retrieve articles that contain a variant of astrobiology, exo-life, or extraterrestrial life and that also contain one or more keywords related to the marine environment. To include all variants of the desired keywords, the wildcard symbol * was used. Specifically, publications were selected using words and word strings in the title (TITLE), abstract (ABS) and keywords (KEY) from articles in journals, congress proceedings, and book chapters. The specific search command was
TITLE-ABS-KEY (astrobiol* OR astro-biol* OR “astro biol*” OR “exo-life” OR “exo life” OR “extraterr* life*”) AND TITLE-ABS-KEY (ocean* OR water OR marine OR “sub ice” OR “sub-ice” OR world* OR moon* OR “water-rock” OR “water rock” OR “water-sediment” OR “water sediment” OR “water world*”OR “Ocean* World*” OR “ocean* moon*” OR “water moon*” OR “water sediment” OR subsurface OR sub-surface OR “sub surface” OR “deep biosphere” OR “deep life” OR “hydrothermal vent*” OR “extreme environment” OR “marine extreme environment*” OR “vent*” OR “seep*”).
where TITLE-ABS-KEY is a combined field that searches titles, abstracts and keywords, AND is a logic operator indicating that the results have to include all words even if they are semantically far apart, OR is a logic operator indicating that articles that contain any of the words are to be retrieved, and * indicates that words of interest include any modification of the stated initial part of a keyword, e.g., astro-biol* contains astro-biology, astro-biological, astro-biologic, astro-biologist, astro-biological investigation, etc.

Overall, 60% of the retrieved articles (randomly selected) were validated by manually checking first titles and then abstracts. Those found to be of no interest because they did not deal with the topics of our search were eliminated. From the retrieved articles, a list of terms was then extracted, by selecting only those that appeared in at least 15 different publications (i.e., in their title, abstract or keywords, not in the body of the article). These terms describe the research topics, modes, and approaches to the research in each article.

At this stage, we created a thesaurus file of the terms where consistency in their spelling was ensured and where only one term was selected among synonyms (Appendix A). The terms that were too general (e.g., attempt, attention, authors, etc.) were eliminated by manual screening (based on the consensus by all the authors of the study) and replaced with AAAA in the thesaurus file. This thesaurus was then used for subsequent analysis.

### 2.2. Analysis of Term Co-Occurrence with VOSviewer

We used the Visualisation of Similarities (VOS) viewer software (Centre for Science and Technology Studies, Leiden University, Netherlands—hereafter referred to as VOSviewer, version 1.6.16; www.vosviewer.com; accessed on 23 February 2023) to create a bibliometric map of the terms, where they appear ranked according to their importance (number of times they were found) and interconnected according to coappearance in the publications. A full explanation of the method can be found in the references [68,74,75]. A Research Information Systems (RIS) file containing the tags in standardized format (developed by RIS Incorporated) to enable citation programs to exchange data was generated by VOSviewer and is included in Appendix A.

The software used the VOS mapping technique to display terms. This technique is closely related to the multidimensional scaling method [76] and involves the use of an intelligent local shift algorithm [77] to identify relations within a network of items. The map is based on the co-occurrence of two terms within an article (in the title, abstract, or keywords), i.e., each of those co-occurring terms are shown on the map and linked by a line. Terms that co-occur frequently are found close to each other in this map, while those which are more weakly related (never co-occur or only co-occur a few times; rarely co-occur separately with a third term; etc.) are found farther apart from each other. Each term is identified by a sphere whose size indicates the number of publications in which the term appears [76,77]. Lines are generated between terms according to the level of their interconnection, with only the more prominent ones being displayed, for clarity. Within the map, some terms become “nodes”, i.e., terms which are at the center of multiple connections. At the next level of interrelation, the map is divided into “clusters” of terms, where some such terms are nodes. A cluster represents a set of closely related terms. Each term or node appears in only one cluster. The total number of clusters is defined by a “resolution parameter”, i.e., a parameter that determines how resolved the analysis is (low resolution generates fewer clusters; high resolution generates more clusters). This parameter is set manually. The resolution parameter was set to 0.9, the value that was found to yield the most appropriate level of detail in the cluster structure for subsequent analysis and discussion.

The map can be viewed in its entirety or parts of the map can be selected for viewing, which allows one to investigate particular areas, particular connections, etc. When conducting these investigations, lines connecting terms not observable in the general map may appear, because at that scale they have higher relative importance. The VOSviewer network files to navigate the maps are available in Appendix A. VOSviewer software allows for highlighting the network of the nearest related terms by clicking on one of them. We used this function to highlight networks of the closest related terms around some relevant nodes and reported them as Appendix A.

### 2.3. Co-Citation Analysis Based on Keywords to Generate Article Clusters, Using CiteSpace

Two concepts were integrated in this analysis, co-citation analysis and clustering. The principle of literature co-citation analysis is based on measuring the similarity between two articles according to the number of times they are cited together in a particular article, in a journal, or by authors. The citation of one article is the number of articles that cite it. Implicit in this analysis is that the citation of an article is a function of the time since its publication and the size of the sample of articles considered [78].

Clustering is the process of dividing a set of physical or abstract objects into several groups. Each generated group is called a cluster. Within clusters, the degree of similarity between elements is higher than the degree of dissimilarity between them. For elements in different clusters, their degree of dissimilarity is higher than their degree of similarity.

CiteSpace (version 5.7.R3—Drexel University, Philadelphia, PA, USA) is a citation analysis software used to identify areas of knowledge (in the most general sense) contained in the scientific literature. CiteSpace performs clustering of articles and keywords using a similarity degree based on article, author, and journal co-citations. We used the same pool of articles from SCOPUS (1963 to 2023) selected previously to carry out the analysis of term co-occurrence with VOSviewer. From the similarity relation between the articles, CiteSpace generated clusters of articles that visualized their interconnection [78]. Each cluster corresponds to an underlying, common theme, such as a trend or area, topic, or line of research [79]. After the clustering is complete, CiteSpace automatically provides a label to each cluster by generating a noun or phrase consisting of a pooling of words from the titles, keywords, and abstracts of articles in that cluster [80]. These labels are created on the basis of the tf-idf selection algorithm [81], which transforms the different forms of a word (and those with the same root) into a single form by selecting a specific word suffix to be attached to the root (for example, “mineral”, “mineralogy” and “mineralogical” could all be represented by “mineralogy”) [82,83]. The clusters are numbered according to size, starting with the largest cluster as #0 and ending with the smallest at #n [79].

Thus, the clusters with their labels provide a visual exploration tool to identify trends in the scientific literature that may not be obvious. Further, CiteSpace can be used to identify historical modifications of scientific trends, such as transient and emerging fields, approaches, concepts, or models. This is carried out with the identification of “citation bursts”. A citation burst is an indicator of a very active area of research. It indicates that a particular publication is associated with a surge of co-citations for any duration of time (within the search range, in this case 1963–2023), from one to multiple years. In other words, a citation burst occurs when a particular publication attracts an extraordinary level of attention from the scientific community during a specified time. If a cluster contains numerous nodes with citation bursts, then the cluster as a whole captures an active research area or emerging trend. Citation bursts in CiteSpace are identified through the Kleinberg’s algorithm, which uses an infinite-state automaton, in which bursts appear naturally as state transitions, producing a nested representation of the set of bursts that imposes a hierarchical structure [84]. After the identification of citation bursts, we used the Citation Burst History function in CiteSpace to generate a historical list of citation bursts labelled with the name of their corresponding cluster (from the clustering analysis of CiteSpace). Each burst was assigned a start and end date, as well as a strength value based on the number of co-citations. The Citation Burst History feature also generates a summary list of articles associated with citation bursts, showing those references causing the strongest citation bursts and the time periods of the strongest bursts [79].

## 3. Results

The query was performed on the 23rd of February 2023 and retrieved 2592 documents with 365 thesaurus terms. The 2022–2023 publications were not yet completely introduced in SCOPUS by the technical staff and a portion of them are not represented in this study. The publications considered in our analysis started from 1963 and ended at the time of the query. The number of retrieved papers remained between 0 and 4 per year until 1997, after which they progressively increased (Figure 1). The main recurring terms found in the titles of the 37 papers published between 1997 and 2000 are related to the following main arguments: Europa (4 items), Mars, and microorganisms (7 and 12 items, respectively, sometimes co-occurring), and finally SETI (3 items).

### 3.1. Emerging Research Trends from Term Co-Occurrence by VOSviewer Analysis

An initial total of 883 terms were obtained from the SCOPUS search considering a minimum of 15 occurrences. The thesaurus (Appendix A) provided consistency between different spellings of terms and synonyms and excluded terms of too general meaning. The final number of terms analyzed with VOSviewer was 365.

The VOS clustering map identified three major groups of terms, as trends in astrobiological research: the search for life on Mars; astrobiology within our Solar System; and astronomical and biological parameters for habitability of planets, including the evolution of life on Earth and exo-planets (Figure 2). The trends and key terms are described below.

“Search for life on Mars” (Color-coded red in Figure 2). This research cluster focuses on two elements: (1) geological and mineral settings with available water and thus compatible with the existence of life; and (2) potential metabolic pathways and ecosystem variables compatible with microorganisms. The terms in this cluster are poorly linked, reflecting more isolated research efforts in comparison with the other two clusters. The terms for the organisms in the upper-left corner (e.g., “prokaryote”, “extremophile”, “halophile”) are associated with “ecology”, “adaptation”, and even “animal”, all grouped together and mostly separated from another group of terms in the lower-left corner related to geology (e.g., “deposit”, “lake” and “subsurface”) and mineralogy (e.g., “mineralogy”, “silica”, “carbonate”, “calcite” and “quartz”), with a small group of words specifically referring to salts and to “salt loving microorganisms” (e.g., “gypsum”, “salt”, “crust”).

“Astrobiology within the Solar System” (Color-coded green in Figure 2). This cluster indicates the large investment in exploration (“exploration”, “mission”) of icy moons as the main candidates to support life (“Enceladus”, “Titan”, “Europa”, “Calisto”, “Ganymede” and even “Pluto”, “ice”, “water ice”, “ice shell”, “polar region”, “thickness”). Also included here are the technological tools for this exploration (e.g., “platforms”, “probes”, “instrumentation”, “sensors”, “mass spectrometry”, “laser”) and geological environment (e.g., “crater”, “geology”, “regolith”). It can be seen in Figure 2 that this cluster is more intertwined with the third cluster “Astronomical and biological parameters for habitability” than it is with the first, “Search for life on Mars”.

“Astronomical and biological parameters for habitability of planets”. This cluster focuses on the evolution of life on Earth and exo-planets (Color-coded blue in Figure 2). The primary research in this cluster is the investigation of those parameters on Earth and within the Solar System to identify potentially life-bearing exo-planets (e.g., “habitability”, “oxygen”). In particular, the detection of planets is accompanied by the analysis of mass and orbital parameters (e.g., “obliquity”, “orbital”, “radius”, “eccentricity” and “tidal heating”), and those of the host star (i.e., “hot star”, “mstar”). In the central and lower part of the cluster, reference to “liquid water” and “ocean” emerges. Interestingly, research on extraterrestrial intelligence and philosophical–societal implications is present in the upper right part, though poorly connected with other terms within this cluster.

Considering the information provided by the analysis of the three clusters together, by focusing on the size and the interconnections of most cited terms, the central role of “life” and “Earth” (the second most recurring term) becomes evident. Other terms, such as “planet”, “environment”, “water”, “Mars”, and “exploration”, also appear as largely co-occurring terms and with numerous interconnections with the other clusters.

### 3.2. Emerging Research Trends form Article Co-Citation Analysis by CiteSpace

The map of research trends (Figure 3) better represents the data-driven studies rather than the concepts or trends involved (Figure 2). CiteSpace allows you to identify groupings, or clusters, using the clustering function [79]. To start the clustering function, simply click on the icon corresponding to the “cluster”. To verify that the clustering process has been completed, simply look in the upper right corner of the drawing area, where the number of clusters will be displayed. Each cluster corresponds to an underlying theme, topic, or line of research. To characterize the nature of an identified cluster, CiteSpace can extract noun phrases from the titles (T in the icon below), keyword lists (K), or abstracts (A) of articles that cited that particular cluster. Once this process is finished, the labels chosen will be displayed. By default, labels based on one of the three selection algorithms, namely tf*idf [79], will be shown. CiteSpace analysis identified ten major clusters from #0 (the largest number of co-citations) to #9, as described below.

Cluster #0 “Yellowknife Bay Gale crater, Mars” groups papers that cite the studies of Hecht et al. [85] and Mumma et al. [86]. Hetch et al. [85] described the existence of significant perchlorates in the Martian soil analyzed by the Phoenix Lander, which points to past hydrological activity. Mumma et al. [86] observed atmospheric methane plumes from Earth-based telescopes, possibly due to microbial metabolism. The label of this cluster refers to Yellowknife Bay, a geological formation in Gale Crater (Mars), explored by NASA’s Mars Science Laboratory (MSL), which landed the Curiosity rover, in 2012. This area is significant because it contains evidence of past hydrological activity on Mars and potentially habitable conditions, including a lake that existed billions of years ago as indicated by petrology and mineralogy, particularly sulphate minerals (associated with evaporated water) [87,88].

Cluster #1 “ice grain” encompasses papers that cite Hsu et al. [40] and Waite et al. [42], dealing with the phenomenon of cryo-volcanism on Enceladus. Also, ice grains are present in the interstellar medium and protoplanetary disks. Articles referring to these studies deal with the presence and behavior of water, and in particular ice, in the interstellar medium.

Cluster #2 “general circulation model” includes papers that most frequently cited Gillon et al. [89] and Kopparapu et al. [90], describing terrestrial-like habitable environments, based on the planet’s atmospheric General Circulation Model (GCM), controlling the temperature, pressure, and chemical composition of the atmosphere. Advances in observational and modelling techniques are allowing researchers to better understand the complexities of planetary atmospheres and their GCPs, providing valuable insights into the potential for life beyond Earth. For instance, Gillon et al. [89] reported the discovery of seven temperate terrestrial planets orbiting the nearby ultracool dwarf star TRAPPIST-1, three of which are located in the star’s habitable zone. The modelling of planets’ atmospheric conditions suggested temperate surfaces with the possibility of hosting liquid water. Kopparapu et al. [90] developed a new method for calculating the habitable zones around main-sequence stars, which considers the effects of clouds, atmospheric chemistry, and GCPs. They found that the habitable zones around M-class red dwarf stars, which are the most common type of stars in the galaxy, may be significantly broader than previously thought. This significantly increases the likelihood of finding habitable exo-planets.

Cluster #3 “Rio Tinto Spain” includes articles that cited Allwood et al. [91] and Poulet et al. [92]. The discovery of stromatolite reefs in the Australian Early Archaean suggested the possibility for microbial life to exist in early Earth environments [91], potentially similar to those of early Mars. The discovery of phyllosilicates by the Mars-Express spacecraft indicates a warmer and wetter climate in the early history of Mars, which could have potentially supported microbial life [92]. Articles on extreme environments refer to Rio Tinto (SW Spain), as an area containing high levels of heavy metals and acidity, which has been intensely investigated as an analogue to the environment of the early Earth or for some areas of early Mars.

Cluster #4 “habitable zone” groups articles that cited Barnes et al. [93] and Westall et al. [94], which discuss the problem of searching for life-bearing planets within the habitable zone of planetary systems. The habitable zone is defined as the region around a star where temperatures are suitable for liquid water to exist on the surface of a planet. The concept of the habitable zone is a crucial tool for astrobiologists in the search for habitable planets. However, as research like Barnes et al. [93] has shown, the habitable zone is a complex and dynamic region that requires careful consideration of a range of factors, since liquid water exists under frozen caps in planets or moons outside the habitable zone in our own Solar System. Furthermore, as Westall et al. [94] demonstrate, astrobiologists need to consider that the very wide range of environmental requirements of microbial life on Earth pushes for setting wider, more flexible habitability criteria.

Cluster #5 “ExoMars rover” mostly groups articles that cited Ojha et al. [95] and Westall et al. [96]. Most of these papers are related to the identification of bio-signatures on Mars [96] and are of direct relevance to the ExoMars rover mission and its instrument package. One intriguing feature of the Martian landscape that the rover could likely encounter is seasonal changes in the soil aspect due to water flux in the warm periods [95], caused by the presence of hydrated salts, which could have important implications for the search for life on Mars. Once on the surface, the rover will conduct a thorough survey of the Martian landscape, collecting and analyzing samples and searching for signs of past or present life.

Cluster #6 “exomoon habitability” includes articles that cited Lammer et al. [97] and McEwen et al. [98]. Lammer et al. [97] considered the key factors that make a planet habitable, including its location in its planetary system, its atmosphere, its magnetic field, and its geology. In considering these factors, the authors also analyze the icy moons of our Solar System as examples of habitability beyond the Goldilocks range. McEwen et al. [98] provide details of seasonal flows on warm slopes on Mars, possibly caused by briny water, as a potential habitability factor. Briny solutions may exist within moons’ external ice shells and are promising places to search for life.

Cluster #7 “unresolved question” groups articles that cited Vago et al. [99], Hays et al. [100] and Goesmann et al. [101], addressing the problem of defining and searching for bio-signatures on Mars and other planets. These challenges are complex and directly affect the ExoMars Rover mission in its search for signs of life [99]. Research methods can be refined in terrestrial Mars analogue environments [100]. Goesmann et al. [101] described how the characterization of organic material in Martian sediments will be carried out using the Mars Organic Molecule Analyzer (MOMA) planned to go on board the ExoMars rover.

Cluster #8 “Venus astrobiology mission design” groups articles that cited Ramirez et al. [102], who revisited the traditional habitable zone concept, which only considers the distance from a star, and introduced new factors such as a planet’s atmospheric composition and the presence of greenhouse gases. In this framework, the still debated presence of phosphine in Venus clouds as a biosignature of microbial life has sparked interest for new missions aimed at searching for Venusian life. Venus could preserve biomarkers of ancient life or even current microbial life in its atmosphere or subsurface, which could be detected by future astrobiological missions. In order to effectively design and execute these missions, it is crucial to have a clear understanding of the Venus habitable zone.

Cluster #9 “environmental state” encompasses articles citing Gillon et al. [89] and Luger et al. [103] that deal with the habitability of planets orbiting M-class red dwarf stars, such as the TRAPPIST-1 system [89]. The TRAPPIST-1 planets are considered promising candidates in the search for extraterrestrial life, although life there is challenged by extreme water loss and its remote detection may be difficult due to the build-up of abiotic O_2_.

The overlap between clusters is rather limited, indicating some common citations between them. There is overlap between clusters #0 and #6, between #7 and both #1 and #8, and between #1 and #6.

The 18 most reported keywords (Figure 4) were largely determined by the publications in the last third of the study period (2005–2023; see Figure 1), a likely result given that the number of publications increases with time. This is not a problem for the study because our main interest is the examination of the present research trends and conceptual connections in astrobiology studies. Thus, the 18 top keywords focus on the most interesting period of our study. This table displays color-coded time ranges to indicate when the keyword was not used, when the keyword was used but it did not burst, and when the keyword burst. In detail, the Citation Burst is an indicator of a very active search area [79]. A burst can last for multiple years, as well as for a single year. A citation burst shows that a particular publication is associated with a wave of citations. In other words, the publication has evidently attracted an extraordinary level of attention from the scientific community. Furthermore, if a cluster contains numerous nodes with strong citation bursts, then the cluster as a whole captures an active research area or an emerging trend. Using the “View—Citation Burst History” feature, a summary list of articles associated with citation bursts can be generated. This visualization shows which references have the strongest citation bursts and over what time periods the strongest bursts occurred [79].

The keywords are a mixture of rather generic terms that cannot be immediately related to research trends (e.g., “instrumentation”, “device”, “controlled study”, etc.) and terms with obvious conceptual and research goal links (e.g., “Martian surface analysis”, “Titan”, and “Enceladus”). Little can be said about the former unless the specific papers are perused, which is beyond the scope of this contribution. With respect to the latter, some comments may be offered. First, the use and bursts of “oceans and sea” and “ocean world” are very recent and overlap, with “oceans and sea” preceding “ocean world” for a few years. Of the four planets or moons mentioned in the keywords, Mars, Titan, and Enceladus have been in use for all or most of the recent period, 2005–2023, although bursts appear at different times. Venus is mentioned only in the two most recent years, albeit intensely, possibly due to recent references to phosphine [104]. It is of interest that the bursts corresponding to Titan and Enceladus are among the longest recorded for non-generic keywords, although their strength is well below that of the “Martian surface analysis”. Overall, the latter is the most cited among the non-generic keywords, considering both the time range and strength of the burst. This is consistent with the intensity of Mars exploration since the mid-1990s. Finally, it can be indicated that “microflora” and “microbiota” have been frequent keywords since 2018, an indication that astrobiology studies focus on extinct or extant microorganisms as the most likely discoveries.

## 4. Discussion

We explored the emerging trends in astrobiology research to identify interactions with the field of marine sciences and future synergies expediting the exploration for extraterrestrial life in exo-oceans. We identified three main scientific areas (Figure 2): (1) the search for life on Mars; (2) the search for life in the rest of the Solar System, with reference to icy moons and their exo-oceans; and (3) the astronomical and biological conditions potentially allowing life on extraterrestrial planets, which includes life evolution and reference to oceanic masses. These three major themes appear to explain the different citation clusters provided by CiteSpace, but with different nuances (see Figure 3).

In our analysis, there are connections between astrobiology and Earth biology, including indications that efforts made in marine sciences could be linked to developing the search for extraterrestrial life. Due to the specific marine-oriented character of our query, our results regarding emerging astrobiology research are different to some extent from those obtained by Malaterre et al. [71]. The latter authors described the following emerging research topics (as clusters of scientific terms, similar to ours in Figure 2): (1) the survival of microbial organisms in extreme conditions, but with such conditions referring mainly to travel in space; (2) prebiotic planetary chemistry, astrochemistry and origins of life, the topics of which were dispersed among our term clusters; (3) planetary formation and habitability, similar to our blue cluster; and finally (4) geological and other potential biosignatures, an area which would be included in our red cluster on Mars research. The bibliometric study by Aydinoglu et al. [70] in relation to the origins of life displayed a Mars-centered cluster of emergent research, similar to the red cluster in our study.

In our term-map analysis (see Figure 2), the fact that “Earth” appears within the blue cluster probably reflects the fact that the exploration of life beyond our Solar System relies on the need to identify astronomical and biological parameters for the habitability of planets based on what we know for our Solar System and the geological history of Earth itself. This view also appears to be reflected in the CiteSpace keyword analysis because it is found within clusters (see Figure 3) number 2, 4, 6 and 9, i.e., developing understanding of how the atmosphere, magnetic field, and geology affect a planet’s habitability conditions, mainly in relation to the existence of liquid water.

There is a variable degree of connectivity of the “Earth” term with many other terms in the other two clusters (see Figure 2), indicating how terrestrial research is of reference for the exploration of life on Mars and in the Solar System. In particular, the research characteristics emerging from our analysis indicate that Mars studies have acquired a certain degree of independence from Earth, with multiple probes and tools (e.g., rovers) able to investigate specific areas, where the search for life on the surface and underground is one of the main drivers. Much of this investigation does not require reference to the Earth, although envisaged life forms are necessarily based on those on Earth and some important links are made to extremophiles in acidic environments [105]. This is also reflected by CiteSpace’s output (see Figure 3), where two well-structured clusters of citations emerge for “Yellowknife Bay Gale crater, Mars” and “ExoMars rover” (#0 and #5, respectively).

The search for life within the Solar System (green cluster in Figure 2) also has a certain degree of independence from Earth (blue cluster in Figure 2) for the same reason. Probes have been sent, providing enough data to develop investigations with less reference to Earth. This is reflected in the CiteSpace output for the “ice grain” cluster (#1). However, the terms “ocean”, “hydrothermal activity”, “hydrothermal vent”, “Enceladus” and “Europa” [106,107], and even Venus [108,109], all appeared within the clusters of term analysis (although “hydrothermal vent” is not represented in Figure 2 due to its sparse occurrence; see Appendix A). They all indicate a reference to Earth environments as potential analogues of the icy worlds of Enceladus and Europa and of the rocky and cloudy environment of Venus.

At the same time, our results indicate that the astrobiology research on Mars is independent from life-oriented research themes for icy moons. The green-cluster terms of the term-map (see Figure 2) related to icy moons fall at the opposite side of microbial-related terms belonging to the red cluster, with little connection. This indicates two independent trends of research, one dedicated to life conditions in exo-oceans and the other to microbial life in terrestrial habitats supposedly similar to those on Mars [110,111,112].

A more in-depth discussion of elements of connection among the three clusters of Figure 2 can be conducted by varying the visualization of connections between specific terms. The most important term connecting all three clusters is “water”, which is at the center of Figure 2. We visualized the direct connections of “water” (by subduing the other terms and connections, see methods; Appendix A). This term is connected to “ice” in the green cluster and to “liquid water” and “oceans” in the blue cluster, and to all important terms. This centrality and connectivity demonstrate that water remains the single most important requirement for models of extra-terrestrial life [113]. The relevance of water research in association with “Mars” was also previously highlighted [70], although with no reference to terrestrial and extraterrestrial oceans.

One other important term for all three clusters is “biosignature” (Appendix A). Missions seeking bio-signatures in situ on Mars and icy moons of our Solar System are a challenging task because organisms and their metabolic products may be destroyed by the harsh conditions experienced on irradiated planetary and icy-moon surfaces [114,115,116,117,118]. Bio-signature search in distant exo-planets is a rather different task and focuses on the spectral analysis of atmospheres.

### 4.1. The Search for Life on Mars

Our results show how Martian extraterrestrial life is often associated with Earth-based biological concepts. On Mars, part of the term map (see red cluster in Figure 2) refers to hypotheses of microbial evolution in putative habitable environments as similar to those on Earth. Having partially reconstructed the evolution of life in water from Earth, researchers are trying to see how the several evolutionary stages could fit the aquatic Martian environment. Therefore, astrobiologists are looking for extraterrestrial microorganisms similar to those on Earth, with similarities ranging from generic characteristics such as being associated with water [111], having carbon-based metabolism and nucleotide-based genetic information that encodes for amino acids assembled into proteins [70], and having more specific characteristics such as spherical or rod-shaped external membranes and the capability for dispersal and motility [119,120,121,122,123,124].

If the connectivity of the most important terms in the search for life on Mars is highlighted, an interesting plot develops in which the term “Earth” is connected to other terms through “microorganism” (Appendix A). Most investigators agree that primary targets for the search for life on Mars are benign paleo-environments (e.g., abundant water, circum-neutral pH), either on the surface or underground, as manifested by the petrology and mineralogy of the geological environment [125,126,127]. The terms “soil”, “lake”, and “crust” (i.e., meaning planetary crust) primarily suggest such environments (Figure 2). At the same time, the evolution of Mars’ surface towards dryer, saltier, colder, and more acidic conditions in the last 3.5 Ga has provided motivation for numerous research efforts into life in extreme conditions on Earth as proxies for Martian environments [32,111,112,128,129,130]. Salty environments generate uncommon conditions that modify the structure of microbial ecosystems because water movements and rain greatly mobilize salts in patchy distributions with salt “crust” deposited above the soil surface [131]. Hypersaline environments are thus also appropriate as analogues of some of the Martian palaeo-lakes, such as the Spotted hypersaline lake [110], where the community of extremophiles is dominated by Firmicutes, Bacteroidetes, and Proteobacteria.

Near-surface and subsurface areas of Earth are inhabited by extremophiles that might have or have had Martian equivalents [105,132,133,134,135,136]. The interest in extremophiles in the context of Martian exploration is manifested in the strong links of the term “extreme conditions” with “Earth” and other important terms in the search for life (Appendix A). All extremophiles are able to live where most other microorganisms die. Given their adaptability and their biogeographical distribution in every “extreme” environment, they could be potentially found on Mars and beyond. As such, several sites on Earth are of relevance for astrobiology, since they can help us to understand “habitability” (blue cluster in Figure 2) and “biosignature” (see Appendix A). This is clearly visible in CiteSpace analysis (see Figure 3), where the cluster “Rio Tinto” emerges (#3). Rio Tinto (SW Spain) is a popular site for astrobiological research since microbial communities live in acidic environments in this location [137]. Similarly, references in this cluster are given to the Salar de Atacama (Chile), investigated as a dry desert with high levels of UV irradiation and salt concentration [138]. The red stone of the Salar de Atacama Desert displays a substantial occurrence of sulfate-reducing bacteria, cyanobacteria, and other “microorganisms” with unique genetic sequences that are not phylogenetically classifiable and displays a peculiar lipid “biosignature” [112,128]. Also, references to the lake of Poàs Volcano (Costa Rica) appear in the Rio Tinto cluster since the microbial community (mainly dominated by *Acidiphilium* spp.) living in its acid-sulphate hydrothermal systems has been the object of research, revealing peculiar genetic adaptations and functional pathways [111] to create energy in cases of carbon limitations.

### 4.2. The Search for Life on Icy Moons and Their Exo-Oceans

Our data show the presence of a structured research cluster dedicated to missions on icy moons of the Solar System, with reference to polar and deep-sea areas on Earth as test environments for extraterrestrial life habitats within ice shells and in underlying exo-oceans (see green cluster in Figure 2). Research on habitable zones including those beyond Goldilocks orbital ranges, centered on ice, is reflected in CiteSpace analysis outputs (see Figure 3) for the clusters “ice grains” (#0) and “habitable zone” (#4).

This is visible in Appendix A, where the exploration missions to Europa and Enceladus icy moons are connected to “Earth”, “ocean”, and “ice”. Earth glaciers are formed from the gradual and continuous accumulation of snow and store information on past environmental conditions of our planet. Microorganisms inhabiting glaciers provide valuable information on the adaptive strategies of extraterrestrial life potentially inhabiting icy moons [136,139,140,141,142]. Marine hydrothermal and deep-sea habitats are postulated as models for putative life-bearing extreme environments in exo-oceans. Hydrothermal vents were possibly widespread in the oceans at the beginning of life on Earth. Deep ocean and submarine polar regions include a combination of the most extreme conditions for the development of life (high pressures, lack of light, and low temperatures), [64,143,144,145,146,147,148,149], and they also host hydrothermal vents.

The need for the molecular detection of life-associated compounds has motivated different research strategies. In situ tests by both space missions and laboratory trials in simulated space conditions have been performed (e.g., terms “mass spectrometry”, “chemistry”, and “molecule”; Figure 2). For example, the ability of orbiting mass spectrometers to detect degrading biomarkers in icy-moon plumes has been tested by comparing observational molecular data from planetary missions with the results of laboratory degradation of bacteria and organic molecules produced by simulated hydrothermalism and radiation [106,150,151,152]. Also, a catalogue of spectra of ice populated by a range of microorganisms was developed to aid the search for extraterrestrial biosignatures focusing on pigmentation [153].

Interestingly, the terms “Europa” and “Enceladus” are linked with “Mars”. Such links exist in the literature that refer to the highest probability of finding exo-life within the Solar System [154], with common approaches to search for life [155,156], in similar cold environments as models [157], or because of similar technology applicable for life identification [158] in the three planetary bodies. This is also probably reflected in the CiteSpace cluster “exomoon habitability” (#6; Figure 3).

### 4.3. Astronomical and Biological Parameters for Habitability of Planets, Including the Evolution of Life on Earth and Exo-Planets

At the date of the present analyses, 3583 exo-planets have been discovered (https://exoplanetarchive.ipac.caltech.edu/docs/counts_detail.html, accessed on 3 April 2024), and this number is expected to increase faster than ever as new telescopes dedicated to exo-planet search come online [159]. Our results (the blue cluster in Figure 2) show how the search for exo-life in distant exo-planets relies on a concept of habitability built on what we know about Earth and its oceans, as well as on Solar System astronomical data (e.g., specific mass and orbital parameters [160]). Several aspects of habitability research appear as peripheral (i.e., not highly connected) in the blue cluster in Figure 2 (e.g., “water vapor”, “surface temperature”, “plate tectonic” and “tidal heating”). Connected with this, CiteSpace (Figure 3) also shows well-structured clusters of cited research in association with “general circulation model” (#2), “habitable zone” (#4), “exomoon habitability” (#6), and “environmental state” (#9). All of these clusters describe the attempt to introduce new Earth-based parameters into astronomical models, aiming at assessing the chance of life (and its potential evolution) on planets with similar characteristics. This comparison is highlighted in the blue cluster of Figure 2, where “atmosphere” (close to “habitability” and “cloud”) and “ocean” are central themes. “Atmosphere” includes the investigation of whether gas spectral characteristics of exo-planets could be interpreted as bio-signatures, given what we know of geological traces left by the evolution of life on Earth. Life could be detected according to astronomical biosignatures such as specific proportions of atmosphere gases such as “oxygen” and reduced gases such as molecular hydrogen, methane, carbon monoxide, and phosphine [102,161,162,163,164,165,166,167,168].

Gases as bio-signatures may occur as a global planetary trait because life performs broad-scale ecological functions in relation to the transference of organic matter (and embodied energy). Here, astrobiology seems to adopt an ecosystem-based concept for the terrestrial biosphere, where planetary atmospheres (and their bio-signatures) emerge from the sum of all life’s metabolic activities in interaction with geological processes in different marine and land ecosystems. This vision recalls homeostasis theories [169], according to which planetary habitability is more related to the biological regulation of the atmosphere and climate than to the luminosity and distance of the host star.

Conditions able to support life in the subsurface, either deep in the planet’s crust or in a subsurface ocean, might extend the habitable zone around stars far beyond the narrow range defined by the presence of surface liquid water [170]. This is clearly represented by CiteSpace research in the cluster “habitable zone” (#4; see Figure 3). While the presence of life on the surface of the planet can clearly impact the atmospheric composition, allowing for remote detection, the possible effect of a subsurface biosphere on the atmospheric inventory is less clear [171]. It might be possible for extraterrestrial life to impact the composition of the planet’s atmosphere even from the subsurface, as long as there is a way to exchange and recycle volatiles between the planetary surface and the interior. Recent work has demonstrated that, on Earth, microorganisms present in the subsurface might alter the quality and quantity of volatiles exchanged between its interior and the atmosphere, with implications on gas composition [171,172,173,174,175]. Certainly, Enceladus possesses an energetic mechanism of connection between the subsurface ocean and atmosphere through water jet injection into the atmosphere [2]. Ceres shows evidence of meteorite impact breaking through the ice surface and connecting the inner briny layer with the atmosphere, while Europa and Ganymede present evidence of the underlying ocean water mixing with the overlying ice, enabling an intermediated ocean-atmosphere contact [2]. All of these mechanisms may operate in exo-planets with an under-ice ocean.

A sub-cluster (upper right corner) containing research related to the terms “intelligence”, “intelligent life”, “extraterrestrial intelligence”, and “philosophy” also appears as peripheral (i.e., see the poor connections of those elements in the blue cluster in Figure 2). Reference to this research is absent in CiteSpace analysis, consistently with its poor connectivity (Figure 3). These terms indicate the presence of a research effort linked to the analysis of habitability and environments capable of hosting intelligent life [176,177] and how this should be detected [178,179,180,181,182,183]. In some cases, the concept and the definition of intelligence itself need to be investigated and revised (justifying the term “philosophy” in the sub-cluster), in order to perform appropriate searches for extra-terrestrial intelligence [184,185]. In fact, philosophical aspects of the existence of extra-terrestrial life are investigated [186,187,188,189,190], including theological implications [191,192]. Terms related to intelligence could also refer to the artificial intelligence and robotics solutions needed to perform autonomous space missions [193,194,195].

### 4.4. Synthesis of the Three Emerging Research Areas: Life on Earth as Paradigm Constraining Astrobiology and Ocean Research

Overall, our results indicate to what extent astrobiology is conditioned by what we know about life on Earth and its adaptation strategies in extreme environments. Our conception of extraterrestrial life is overwhelmingly carbon-based, while its closest candidate concept, silicon-based life, remains unpopular [196]. This is shown in that the term “silicon” did not even appear in the thesaurus list from which the term-map was generated (see Figure 2). Our vision of life is also linked to solid and liquid water within temperature ranges of extremophile bacteria on Earth. At present, the most important candidates to support life within the Solar System are the icy moons known (Enceladus, Europa, Titan and Ganimede) or assumed (Ceres) to hold under-ice oceans. Their environments share important characteristics with the corresponding habitats on Earth, which makes ocean research a relevant discipline to astrobiological exploration for the development of concepts, models, and exploration tools. The discovery of icy moons’ potential has greatly widened the range of habitable areas in other planetary systems and has made ocean exploration relevant to these systems. These results justify the title of the present contribution.

Also highlighted in our study is the presence of some peripheral ‘salt-related’ terms, pointing to a very relevant topic from an Earth ecological perspective [197] that should also be applicable to extraterrestrial life. Our data indicate that despite the research effort dedicated to the characterization of the microbial features that would allow species to survive in extreme extraterrestrial environments, ecosystem-oriented research focusing on how species interact in extreme terrestrial environments (as proxy for Mars, for example) is still in its infancy. Following Cockell et al. [198], at the most fundamental level, habitability is binary (i.e., at any given time, an environment is either habitable or not for any given organism), while at a superior level, habitability becomes a spectrum of possibilities resulting from the integration of ecological questions such as coexistence and interactions of organisms, all of which build up ecosystem dynamics. Therefore, once the basic questions regarding habitable conditions are established, research efforts should focus on the genetic properties and metabolic pathways of bacterial species that may provide functional flexibility that allows them to interact and survive in extreme environments. Understanding the genetic and metabolic features of extremophiles living in Mars-like environments is crucial in astrobiology [127], as they shed light on the selection mechanisms exerted by the environments on bacteria and archaea groups with functional properties to survive in extreme environments. Microbial life on Earth in such environments is active at high pressures, at temperatures ranging from boiling to freezing, in critical osmotic potentials, in radioactive environments, and in environments with a wide range of pH values and toxic concentrations of metals and other species [113]. If life exists on Mars and other planets, microorganisms must cope with the above challenges and energy limitations and be endowed with numerous metabolic pathways. Each extremophile has its own set of genes and functions to face such challenges [128,199]. At the same time, their metabolites could be beneficial for other species and lateral gene transfer could take place, generating a well-equipped, collaborative, and flourishing ecosystem. In this respect, the full potential and extent of ecosystem collaboration on Earth still need to be discovered and the results need to be projected into a wider and more complete framework for astrobiological research. On Earth, extremophiles have very specific features because they inhabit infrequent environments, but these features need not to be considered a limitation but simply represent the adaptation to specific conditions. On a planet where what we call extreme conditions are the norm, such a designation would not be meaningful.

Extraterrestrial life has been contemplated as inhabiting energy-providing rocky-aqueous environments and being capable of carrying out metabolic functions that allow growth and movement [200,201,202,203,204,205], such as prokaryotic-like life forms (either Bacteria and Archea or protozoans) [124,206]. Also, models for encoding and transmitting information regarding structural character and metabolic activity are based on DNA or RNA, molecules with a series of basic elements arranged in different sequences. Astrobiology is therefore looking for models of extraterrestrial organisms similar to those on Earth. Arguably, one should rather assume that only some level of metabolic and morphological resemblance would exist between terrestrial and extraterrestrial organisms inhabiting similar environments (e.g., Earth deep-sea and exo-oceans or extreme terrestrial and Martian environments), because they evolved independently. The assumption that life anywhere in the universe is based on the same requisites and models as those on Earth is a strong assumption, which has been consistently maintained throughout the entire body of literature analyzed in the present work. The explanation for this perhaps unreasonable stubbornness is that we cannot even imagine any other models to work with.

On an interesting note, our results also indicate the less prominent exploration of multicellular or intelligent (self-conscious) life in astrobiological research. One can assume that biogeochemical and molecular-based methods can detect the potential presence of biomolecules of both unicellular and multicellular organisms. However, if life was discovered, the progress from exploration and life detection to life monitoring would require instruments capable of providing information on the behavioral aspects of the discovered life forms on a macroscopic level (i.e., hyperspectral imaging, bioluminescence sensors, and acoustics; [49,61]). Because unicellular life is considered a much more likely possibility than multicellular life, research priorities and the allocation of resources focus on the former and leave out instrumentation optimized for the exploration for potential multicellular life. Beyond this reasonable argument, the poor emphasis on exploration for potentially intelligent life, which is such a transcendental question for humanity [184,207,208,209], suggests a yet underdeveloped cross-sectoral dialogue with other disciplines such as theology, sociology, and philosophy. However, it appears that philosophers, social scientists, and theologians publish astrobiology studies related to the origin of life [70], but the possibilities and implications for extraterrestrial intelligence are not explored to a similar extent. Lack of attention to putative extraterrestrial intelligence appears to be a common element across the entire range of disciplines.

## 5. Perspectives

We explored the emerging trends in astrobiology research and identified three areas within which interconnection of the bibliography is more intense: life on Mars, life in the rest of the Solar System (with icy moons and their exo-oceans), and the conditions allowing life on extraterrestrial planets. Mars, Solar System, and icy moon studies show a certain degree of independence from Earth-based studies because the space missions provide the data to develop planetary investigations. Nonetheless, the search for life on Mars is entirely dependent on terrestrial life models, and substantially on those of terrestrial extremophiles. Similarly, the search for life in exo-oceans is greatly dependent on terrestrial models. This dependance is reasonable because life on Earth is the only life we know of and a closer synergy between marine and planetary exploration is very likely to help develop successful biological concepts and exploration tools for both disciplines. However, it is also necessary to ask the question of whether the life on Earth paradigm is constraining astrobiology and hindering wider research avenues.

Our study manifests a generalized lack of consideration of ecological relationships within any evolving biosphere. The use of gases as planetary bio-signatures is established, but references to conceptual research on other potential attributes of extra-terrestrial ecosystems, assuming the universal paradigm of evolution via natural selection, is scant. Any extra-terrestrial ecosystem must respond to the limiting energy criteria with uni- or multicellular species organized as primary producers that incorporate stellar and/or geochemical energy in their biomass (as the first step towards the syntropy of several living organisms), sustaining the rest of the food chain. This energy–biomass loop is likely to be closed by decomposers and remineralizing species. In this framework, future avenues of research could use artificial intelligence to model putative extraterrestrial uni- and multicellular organisms living in the environmental conditions occurring on different icy moons and planets, simulate how biomasses and conditions would evolve, and attempt to derive biosignatures that may be identified in telescopic and mission exploration.

## Figures and Tables

**Figure 1 life-14-00676-f001:**
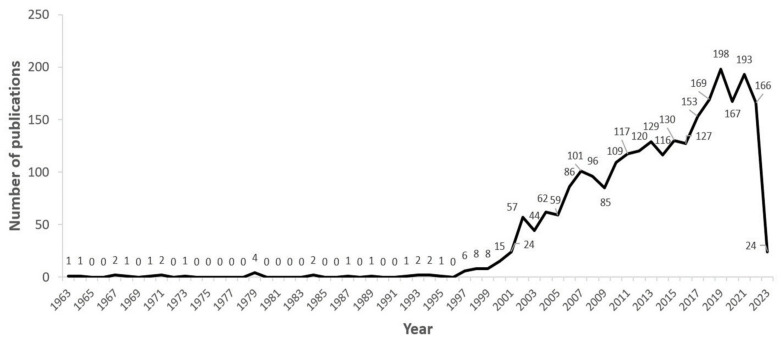
Publication trend in astrobiology research, as represented by the total number of articles identified by our query per year. The 2023 drop in the number of published items results from the query being finalized prior to that year ending.

**Figure 2 life-14-00676-f002:**
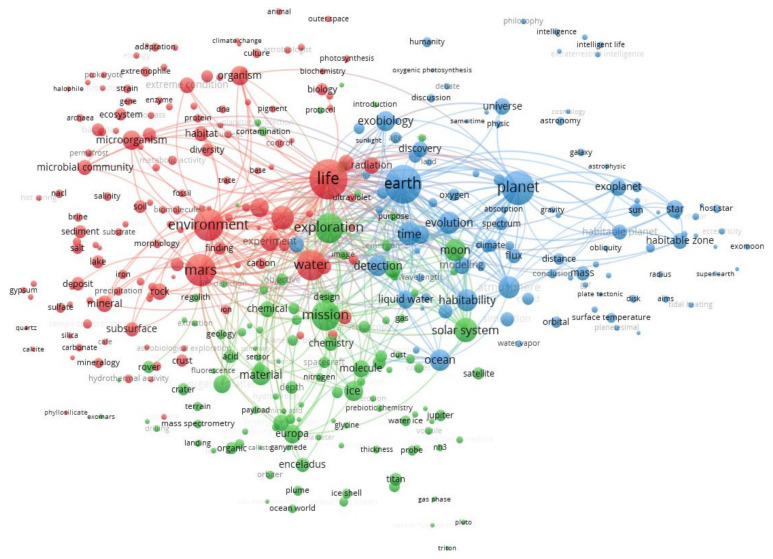
Term-map analysis results by VOSviewer depicting three major emerging clusters in astrobiology research with different levels of interconnection among terms. The top 300 linkages are represented. A term’s importance is indicated by the size of the circle.

**Figure 3 life-14-00676-f003:**
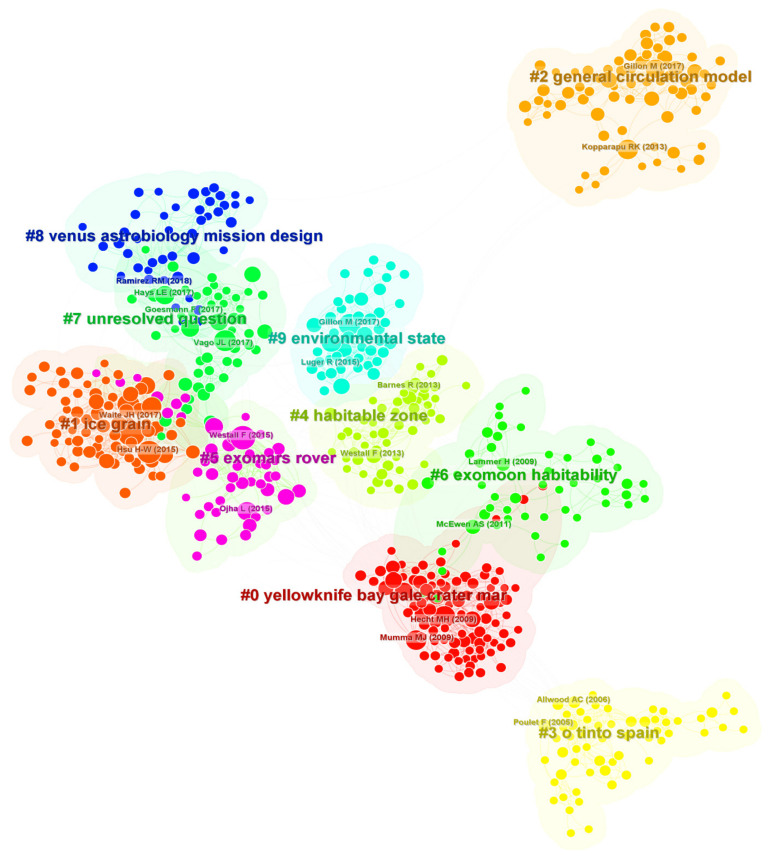
CiteSpace map of research trends. The number and name of each group are assigned based on the quantity of references to selected articles and the articles’ contents.

**Figure 4 life-14-00676-f004:**
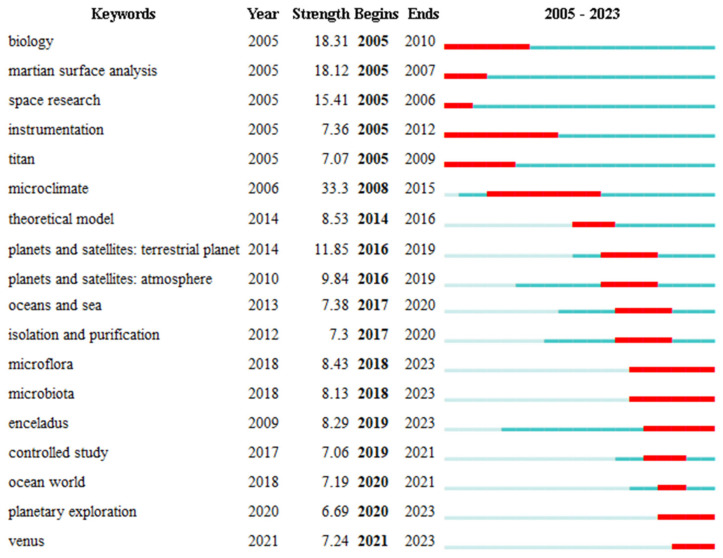
CiteSpace analysis of keywords within papers quoted in research articles retrieved by the query. The colour bars on the right represent the following period ranges. Red: burst in the use of the keyword; dark blue-green: the keyword was used below the burst threshold; pale blue-green: the keyword was not used. “Begins” and “Ends” represent the time interval (i.e., “Year”) when each keyword was used.

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
