# Peer review of "Marine Science Can Contribute to the Search for Extra-Terrestrial Life"

_life, 2024, doi:10.3390/life14060676_

Round 1

Reviewer 1 Report

Comments and Suggestions for Authors

Rather than just a literature search, some component of perspective or original research is needed.

Author Response

Reviewer no. 1

This reviewer finds that our work is “A lot of work has been done to review existing papers. But, the results are trivial and do not give anything on the merits of the problem- namely, origin of life. There is a lack of substantive analysis and scientific conclusions. The work is purely bibliographic and is suitable for another journal.

We acknowledge the reviewer comment about the review effort done, and we feel sorry for the disagreement on the derived merits, especially in relation to the lack of scientific conclusions. Unfortunately, in absence of more precise indications, we cannot further proceed to additional changes in this section of our reply. We tried to improve that aspect by carefully considering the comments by the other two reviewers (e.g., by adding the Conclusion Section; see our reply to reviewer no. 2).

Reviewer 2 Report

Comments and Suggestions for Authors

A lot of work has been done to review existing papers.But, the results are trivial and do not give anything on the merits of the problem- namely, origin of life. There is a lack of substantive analysis and scientific conclusions. The work is purely bibliographic and is suitable for another journal.

Author Response

Reviewer no. 2

This reviewer finds that our work is “A lot of work has been done to review existing papers. But, the results are trivial and do not give anything on the merits of the problem- namely, origin of life. There is a lack of substantive analysis and scientific conclusions. The work is purely bibliographic and is suitable for another journal.

We acknowledge the reviewer comment about the review effort done, and we feel sorry for the disagreement on the derived merits, especially in relation to the lack of scientific conclusions. Unfortunately, in absence of more precise indications, we cannot further proceed to additional changes in this section of our reply. We tried to improve that aspect by carefully considering the comments by the other two reviewers (e.g., by adding the Conclusion Section; see our reply to reviewer no. 2).

Reviewer 3 Report

Comments and Suggestions for Authors

It is a meaningful work to search for extraterrestrial life in the exo-oceans. In this paper, the authors conducted a data-driven bibliographic survey on astrobiology literature to identify emerging research trends with marine sciences for future synergies in the exploration for extraterrestrial life in exo-oceans. Based on search queries, they identified 2592 published items since 1963 and ascribed them into three major groups. They also identified that the most prominent research keywords form three key-groups centred on water-based oceans.  My main comments are as follows.

1.The authors give a detailed description about the work to search for extraterrestrial life in the exo-oceans. I recommend to summarize the questions in current work and to present the possible researches in future.

2.Line 123: change “CiteSpace)” to “CiteSpace”.

3.Line 143-144: The search methodological approaches are important in this work, but the authors only mentioned that ”following similar methodological approaches previously tuned 143 in Costa et al. [73].” I recommend to give a brief introduction to the approaches.

4.Figure 1: “The query was performed on the 23rd February 2023”, thus, the number 24 can not represent the publications in 2023, which can not be compared with other years.

5.Some notations are not clear in Figures 2, 3, 4, and S1, S2, S3, S4, S5.

6.I suggest to add a “Conclusion” Section.

Comments on the Quality of English Language

Minor editing of English language is required.

Author Response

In disagreement with reviewer No. 1, this reviewer thinks that our work “is a meaningful work to search for extraterrestrial life in the exo-oceans”, because we “…. conducted a data-driven bibliographic survey on astrobiology literature to identify emerging research trends…”. The reviewer advises making “minor changes” in the MS.

  1. The authors give a detailed description about the work to search for extraterrestrial life in the exo-oceans. I recommend to summarize the questions in current work and to present the possible researches in future.

We propose to call the last section Perspectives. The reason is that the section does not really contain the conclusions of the paper, rather it makes a summary of the bibliometric results and then further discusses several issues from the discussion, taking them one step further. Perspectives is a more flexible term and goes well with the looking-into-the-future style of the section.

The reviewer’s specific comment has been addressed by introducing a Perspectives section (see also our reply to point no. 6 of this reviewer, below), which now reads:

“We explored the emerging trends in astrobiology research and identified three areas within which interconnection of the bibliography is more intense:  life on Mars, life in the rest of the Solar System (with icy-moons and their exo-oceans) and the conditions allowing life on extraterrestrial planets. Mars and Solar System and icy-moons studies show a certain degree of independence from Earth-based studies because the space missions provide the data to develop planetary investigations. Nonetheless, the search for life on Mars is entirely dependent on terrestrial life models, and substantially on those of terrestrial extremophiles. Similarly, the life quest in exo-oceans is greatly dependent on terrestrial models Such dependance is reasonable because life on Earth is the only life we know and closer synergy between marine and planetary exploration is very likely to help develop successful biological concepts and exploration tools for both disciplines. However, it is also necessary to set the question whether the life on Earth paradigm is constraining astrobiology and hindering wider research avenues.

Our study manifests a generalized lack of consideration of ecological relationships within any evolving biosphere. The use of gases as planetary bio-signatures is established, but references to conceptual research on other potential attributes of extra-terrestrial eco-systems, assuming the universal paradigm of evolution via natural selection, is scant. Any extra-terrestrial ecosystem must respond to the limiting energy criteria with uni- or multicellular species organized as primary producers that incorporate stellar and/or geo-chemical energy in their biomass (as the first step towards syntropy of several living organisms), sustaining the rest of the food chain. This energy-biomass loop is likely to be closed by decomposers and remineralizing species. In this framework, future avenues of research could use Artificial Intelligence to model putative extraterrestrial uni- and multi-cellular organisms living in the environmental conditions occurring in different icy-moons and planets, simulate how biomasses and conditions would evolve and at-tempt to derive biosignatures that may be identified in telescopic and mission exploration.”

  1. Line 123: change “CiteSpace)” to “CiteSpace”.

That was changed.

  1. Line 143-144: The search methodological approaches are important in this work, but the authors only mentioned that “following similar methodological approaches previously tuned in Costa et al. [73].” I recommend to give a brief introduction to the approaches.

As suggested, we added this part: Lines 148-154: “Although new statistical approaches exist to monitor the overall status of various research topics, one emerging method is scientific mapping. This approach allows for a review of research and is designed to synthesize patterns of knowledge production within a discipline, as opposed to synthesizing substantive findings. It is an interdisciplinary field emerging from traditional library information science in the areas of scientometrics, citation analysis, and computer science in the subareas of information visualization, visual analysis, data mining, and knowledge discovery [73].”

  1. Figure 1: “The query was performed on the 23rd February 2023”, thus, the number 24 can not represent the publications in 2023, which can not be compared with other years.

We understand this point, but we think best to keep this year in Fig. 1 for the sake of clarity and consistency with what we describe in the extracted publications data set. Our analysis is not carrying out comparisons between years but using the entire database together. Figure 1 is clear while eliminating the year 2023 would result in a figure not fully representing the data set in its integrity.

  1. Some notations are not clear in Figures 2, 3, 4, and S1, S2, S3, S4, S5.

The reviewer is correct and we embedded a new version of those figures of enhanced quality.

  1. I suggest to add a “Conclusion” Section.

This was added, as indicated above, in reply to a previous comment of this same reviewer but this addition was also motivated by the general comment of reviewer no. 1 (to increase the general value of our research presented in this MS).

Reviewer 4 Report

Comments and Suggestions for Authors

This paper provides an interesting and informative window into the current state of astrobiological research and speculation. It makes use of sophisticated bibliographic tools that identify connections and relationships between different nodes of interest and attention. It actually goes well beyond the limits suggested by the title, depicting investigations not limited to astrobiological projects solely relating marine science on Earth to exo-marine habitats.

The findings of this broad survey of the astrobiological literature for the most part are not surprising.  That presumably arises from the bias inherent in the terms selected by the authors to generate the set of publications to be considered. Nonetheles, it is useful for the astrobiological community to see where the strongest relationships and current trends lie. Of equal or greater value, however, is the way this survey points to relationships and areas that either are not a current focus but should be, or that suffer from a lack of imagination.

The last three paragraphs of the Discussion are particularly valuable. Reference to the “unreasonable stubbornness” of searching for life as we know it is a good point. Actually, a number of authors have shown a greater breadth of imagination in considering other forms of energy, alternatives to water as solvents, and other ways in which alien forms of life could exist under alien conditions, though those publications are not captured by the strategy used in this survey.

One major failure of contemporary astrobiological thinking is the near total lack of consideration of ecological relationships within a diverse evolving biosphere. In the third to last paragraph, the authors appear poised to address the issue of interactions of organisms and the ecosystem dynamics resulting from those interactions, but then they veer off toward a discussion of metabolic pathways and adaptations.  Those are valid subjects, but so are population structures, food webs, biomass, and other attributes of diverse ecosystems (which must inevitably arise under some conditions), yet they are not represented in this survey, and for a reason – very few astrobiologists pay any attention to them.  This is not a criticism of the survey, as it accurately reflects the negligence of these subjects by the contemporary astrobiological community.

While this paper is a well-conceived, nicely executed, and valuable picture of the field of astrobiology at the present time, I experienced a few frustrations with it, at least the way it appears on my computer:

1. The sophisticated bibliographic tools used to generate the figures, VOSview and CiteSpace, apparently depend on inserting datasets into downloadable software.  The Supplementary Materials contain some or all of those datasets, but the authors do not explain how to proceed in a stepwise fashion to generate the figures.  The overall message comes through, but if the datasets are intended to be used, a more detailed explanation of how to do so would be helpful.

2.  The resolution of Figures S1-S5 is very poor on my computer. I can’t even make out some of the critical terms.

3.  I can’t figure out how to access the Supplementary Figures.  All I get are text files when I click on the links.

4. Line 172: What were the criteria for eliminating articles on the basis that they were “those of low interest”?

5. It isn’t clear how or why the terms in Fig. 2 were segregated into three different populations.  The figure is interpreted to highlight “search for life on Mars” (red), “astrobiology within the solar system” (green), and “astronomical and biological parameters for habitability” (blue). Looking at Fig. 2 as a naïve observer, why couldn’t the categories be “life” (red, but not very useful), “mission” (green), and “earth” (blue)? What would be lost if all the clusters and lines were the same color?

6. It was a little surprising that “exobiology” was not used as a term for capturing the original population of articles to be considered, as this would have picked up a lot of the older (especially European) literature, though the term does show up in the thesaurus and in Fig. 2

7. In general, the paper is well written, but in a few instances, the wording and grammar will sound at best awkward and at worst incorrect to native English speakers. The following are a few examples:

Lines 124-5: “could be used to uncovering the evolving nuances”

Line 179: “the too general terms” (instead of “the terms that were too general”)

Lines 241-2: “which allows to convert”

Lines 277-8: “The Thesaurus . . . allowed to ensure consistency”

Line 459: “in order to evidence possible interactions”

Line 683: “environments capable to host intelligent life”

Line 738: “capable to metabolise for movement and grow”

Incomplete sentences in lines 187-8 and 304-5.

Comments on the Quality of English Language

Overall quality is fine, but see #7 above

Author Response

The reviewer thinks in agreement with reviewer no. 2 (and not agreeing with reviewer no. 1) that our work “…. provides an interesting and informative window into the current state of astrobiological research…” through the “…. use of sophisticated bibliographic tools….”, going “…well beyond the limits suggested by the title….”. The reviewer also judges that our “….  findings …… are not surprising…”, because of the “…. bias inherent in the terms selected by the authors to generate the set of publications to be considered.” Notwithstanding, our paper contents are deemed as “…. useful for the astrobiological community to see where the strongest relationships and current trends lie. Of equal or greater value, however, is the way this survey points to relationships and areas that either are not a current focus but should be, or that suffer from a lack of imagination.”

Some gaps in the ongoing search for life have been detected in the “… last three paragraphs of the Discussion”, which “… are particularly valuable …”. One of them is the “Reference to the “unreasonable stubbornness” of searching for life as we know it is a good point”. The reviewer indicates that a number of “… authors have shown a greater breadth of imagination in considering other forms of energy, alternatives to water as solvents, and other ways in which alien forms of life could exist….” and have produced publications “…. not captured by the strategy used in this survey”.

A mention to this failure of imagination in research was briefly introduced in the new Conclusion section (see also our reply above and to Re. no 2), because already specified at Line s736-740 (Discussion Section) a mention to “silicon” life was already present with a referencing study. If more publication should be analysed and inserted there, we would be very happy to read and consider them and we kindly ask to the reviewer to indicate those items to us.

The reviewer also points out that “One major failure of contemporary astrobiological thinking is the near total lack of consideration of ecological relationships within a diverse evolving biosphere. In the third to last paragraph, the authors appear poised to address the issue of interactions of organisms and the ecosystem dynamics resulting from those interactions, but then they veer off toward a discussion of metabolic pathways and adaptations. Those are valid subjects, but so are population structures, food webs, biomass, and other attributes of diverse ecosystems (which must inevitably arise under some conditions), yet they are not represented in this survey, and for a reason – very few astrobiologists pay any attention to them. This is not a criticism of the survey, as it accurately reflects the negligence of these subjects by the contemporary astrobiological community.

We agree on this particular point and we added in a new Perspectives section, a general mention to ecosystem functioning and future avenues of research in a framework that assumes the universal paradigm of evolution via natural selection based on thermodynamic disequilibrium (see our previous comment to reviewer no. 2 about the introduced Perspectives section).

The reviewer also addressed a series of points in the following manner “While this paper is a well-conceived, nicely executed, and valuable picture of the field of astrobiology at the present time, I experienced a few frustrations with it, at least the way it appears on my computer”. All those points have been addressed below, one by one as follows.

  1. The sophisticated bibliographic tools used to generate the figures, VOSview and CiteSpace, apparently depend on inserting datasets into downloadable software. The Supplementary Materials contain some or all of those datasets, but the authors do not explain how to proceed in a stepwise fashion to generate the figures. The overall message comes through, but if the datasets are intended to be used, a more detailed explanation of how to do so would be helpful.

We appreciate this point, and we agree with the reviewer that VOSviewer should not appear as a black box in the article. However, VOSviewer is an easy-to-use free software available at https://www.vosviewer.com. All the documentation is available on the same site. The supplementary files (from 1 to 4) could be easily used to generate the figures and the information reported in the article. The same RIS file (Supplementary material 2) could be also used with CiteSpace. We do not think that inserting a detailed description of the functioning operations in the supplementary materials will be helpful.

Additional information to generate CiteSpace figures has been added as following:

- Lines 382-391: “CiteSpace allows you to identify groupings, or clusters, using the clustering function [79]. To start the clustering function, simply click on the icon corresponding to the “cluster.” To verify that the clustering process has been completed simply look in the upper right corner of the drawing area where the number of clusters will be displayed. Each cluster corresponds to an underlying theme, topic or line of research. To characterize the nature of an identified cluster, CiteSpace can extract noun phrases from the titles (T in the icon below), keyword lists (K), or abstracts (A) of articles that cited that particular cluster.  Once the process is finished, the labels chosen will be displayed. By default, labels based on one of the three selection algorithms, namely tf*idf [79], will be shown.

Lines 507-516: “In detail, the Citation Burst is an indicator of a very active search area [79]. The burst can last for multiple years as well as for a single year. A citation burst shows that a particular publication is associated with a wave of citations. In other words, the publication has evidently attracted an extraordinary level of attention from the scientific community. Furthermore, if a cluster contains numerous nodes with strong citation bursts, then the cluster as a whole captures an active research area or an emerging trend. Using the “View - Citation Burst History” feature, a summary list of articles associated with citation bursts can be generated. This visualization shows which references have the strongest citation bursts and over what time periods the strongest bursts occurred [79].

2.The resolution of Figures S1-S5 is very poor on my computer. I can’t even make out some of the critical terms.

We do not know what may have caused this poor resolution. The resolution of the uploaded figures was high enough for enlargements that allowed to see most of the labels. We wonder if the reviewer did not try to enlarge the figures. However, not all labels for all displayed spheres are present; only sphears above a threshold bear a label to avoid overcrowding, which would prevent any reading).

3.I can’t figure out how to access the Supplementary Figures. All I get are text files when I click on the links.

All the supplementary figures were all embedded in the body of the MS. Also, all our Supplementary Figures were uploaded as separated. We strictly followed all the instructions provided by the journal website. We do apology but we cannot say more on this issue. We advise to ask for assistance to the Editor Staff on that particular question.

  1. Line 172: What were the criteria for eliminating articles on the basis that they were “those of low interest”?

Articles were present that although they contained the keywords of the search, they dealt with topics totally unrelated to the search performed. They were read by the authors of the paper and subsequently discarded because they were not congruent.

  1. It isn’t clear how or why the terms in Fig. 2 were segregated into three different populations. The figure is interpreted to highlight “search for life on Mars” (red), “astrobiology within the solar system” (green), and “astronomical and biological parameters for habitability” (blue). Looking at Fig. 2 as a naïve observer, why couldn’t the categories be “life” (red, but not very useful), “mission” (green), and “earth” (blue)? What would be lost if all the clusters and lines were the same colour?

As written in M&M section (lines 216-240): “The software uses the VOS mapping technique to display terms. This technique is closely related to the multidimensional scaling method and involves the use of an intelligent local shift algorithm to identify relations within a network of items. The map is based on the co-occurrence of two terms within an article (in title, abstract, or keywords); i.e., each of those co-occurring terms are shown on the map and linked by a line. Terms that co-occur frequently are found close to each other in this map, while those which are more weakly related (never or few times co-occur; rarely co-occur separately with a third term; etc.) are found farther apart from each other. Each term is identified by a sphere whose size indicates the number of publications in which the term appears Lines are generated between terms according to the level of their interconnection, with only the more prominent ones being displayed, for clarity. Within the map, some terms become “nodes”, i.e., terms which are at the centre of multiple connections. At the next level of interrelation, the map is divided into “clusters” of terms, where some such terms are nodes. A cluster represents a set of closely related terms. Each term or node appears in only one cluster. The total number of clusters is defined by a "resolution parameter", i.e., a parameter that determines how re-solved the analysis is (low resolution generates fewer clusters; high resolution generates more clusters). This parameter is set manually. The resolution parameter was set to 0.9, the value that was found to yield the most appropriate level of detail in the cluster structure for subsequent analysis and discussion.”

In other words, the tool produces the map, which contains embedded the level of interrelation between terms. There are many possible divisions in clusters which correspond only to the level of cluster resolution chosen by the analyst. We selected that level of resolution that produced the most meaningful result: clearly identifiable clusters that contain a set of terms of recognizable interrelation (we chose the result where these two properties had maximum values). The name of the clusters has been given interpreting the terms included within each cluster. Other names are possible but contain less information, or less specific, about the meaning of the cluster. Having only one cluster is also possible (minimum level of resolution) but such representation misses information contained within the map about how the terms are interrelated and the intensity of that interrelation.

  1. It was a little surprising that “exobiology” was not used as a term for capturing the original population of articles to be considered, as this would have picked up a lot of the older (especially European) literature, though the term does show up in the thesaurus and in Fig. 2.

The term exobiology has not been used in the search commend but, as evidenced by the reviewer, clearly emerged as important term. In fact, it as been located in the blue cluster occurring 297 times. The term is present in the Thesaurus.

  1. In general, the paper is well written, but in a few instances, the wording and grammar will sound at best awkward and at worst incorrect to native English speakers. The following are a few examples:

As indicated all those issues were dealt.

Lines 124-5: “could be used to uncovering the evolving nuances”

That was changed for “…..and analyzing large volumes of scientific data [69], manifesting astrobiology research fields and shedding light on emerging areas….

Line 179: “the too general terms” (instead of “the terms that were too general”)

That was changed.

Lines 241-2: “which allows to convert”

Thnt was changed as “….tf-idf selection algorithm [81], which transforms the different forms of a word…..

Lines 277-8: “The Thesaurus . . . allowed to ensure consistency”

That was changed as “The Thesaurus (Supplementary Information 1) provided consistency…………

Line 459: “in order to evidence possible interactions”

That phrase was simplified ad “We explored the emerging trends in astrobiology research to evidence interactions with the field of marine sciences…..

Line 683: “environments capable to host intelligent life”

That phrase was also slightly simplified: “…. research effort linked to the analysis of environments habitability and intelligent life…

Line 738: “capable to metabolise for movement and grow”

We modified as follows: “Extraterrestrial life has been considered capable to metabolize for movement and grow in rocky-aqueous environments…..

Incomplete sentences in lines 187-8 and 304-5.

The first incomplete phrase was emended as follows: “…..explanation of the method, see the following papers: [68,74,75]

For the other incomplete phrase, it refers to the title of the clusters identified in figure 2. All clusters’ names (at the beginning of their respective paragraph for description) apparat therefore as incomplete (no verbs), but because of being titles we judged they can remain as they are.

Round 2

Reviewer 2 Report

Comments and Suggestions for Authors

After the correction, the potential useful applications of the used search became better visible. A lot of bibliographic work has been done to analyze the links between various fields of research in the life (terrestrial and extraterrestrial) sciences. And although the work itself does not add anything new to the existing knowledge, it can serve as a good review for a scientist who is started to the study of this difficult problem - how and where to explore different forms of life.

Reviewer 3 Report

Comments and Suggestions for Authors

The author have carefully responded to the comments and suggestions. The goal of the paper and the presentation of the topics are much clearer now. My opinion is that the revised version of the paper is worth of publication.